# Gut Microbiota Interventions to Retain Residual Kidney Function

**DOI:** 10.3390/toxins15080499

**Published:** 2023-08-11

**Authors:** Denise Mafra, Julie A. Kemp, Natalia A. Borges, Michelle Wong, Peter Stenvinkel

**Affiliations:** 1Graduate Program in Nutrition Sciences, Fluminense Federal University (UFF), Niteroi 24020-140, Brazil; kempjulie@gmail.com; 2Graduate Program in Biological Sciences—Physiology, Federal University of Rio de Janeiro (UFRJ), Rio de Janeiro 21941-902, Brazil; 3Institute of Nutrition, Rio de Janeiro State University (UERJ), Rio de Janeiro 20550-170, Brazil; nat_borges_@hotmail.com; 4Division of Nephrology, Department of Medicine, University of British Columbia, Vancouver, BC V6T 1Z1, Canada; michellemywong@gmail.com; 5Karolinska Institutet Innovations, 171 65 Stockholm, Sweden; peter.stenvinkel@ki.se

**Keywords:** chronic kidney disease, gut microbiota, residual kidney function, nutrition, interventions

## Abstract

Residual kidney function for patients with chronic kidney disease (CKD) is associated with better quality of life and outcome; thus, strategies should be implemented to preserve kidney function. Among the multiple causes that promote kidney damage, gut dysbiosis due to increased uremic toxin production and endotoxemia need attention. Several strategies have been proposed to modulate the gut microbiota in these patients, and diet has gained increasing attention in recent years since it is the primary driver of gut dysbiosis. In addition, medications and faecal transplantation may be valid strategies. Modifying gut microbiota composition may mitigate chronic kidney damage and preserve residual kidney function. Although various studies have shown the influential role of diet in modulating gut microbiota composition, the effects of this modulation on residual kidney function remain limited. This review discusses the role of gut microbiota metabolism on residual kidney function and vice versa and how we could preserve the residual kidney function by modulating the gut microbiota balance.

## 1. Introduction 

Gut microbiota disturbances have been reported in patients with chronic kidney disease (CKD) in all disease stages. Although most studies have been conducted in haemodialysis patients, a few have been conducted in patients on peritoneal dialysis and patients not on dialysis [1]. In general, these studies show reduced alpha diversity and an accumulation of uremic toxins generated by the gut microbiota, such as phenylacetylglutamine (PAGln), cinnamoylglycine, p-cresyl sulfate (p-CS), phenyl sulfate (PS), hippuric acid, indoxyl sulfate (IS), p-cresyl glucuronide, trimethylamine N-oxide (TMAO), indol acetic acid and phenylacetylglycine (PAGly). Moreover, studies have shown that levels of short-chain fatty acids (SCFA) produced by certain bacteria (*Blautia*, *unidentified Ruminococcaceae* and *Parabacteroidies*) with beneficial effects on health (anti-inflammatory responses, gut barrier modulation and reduction of bacterial translocation) are reduced because of limited SCFA-producing bacteria [1,2]. Deleterious effects, such as inflammation, oxidative stress, atherosclerosis and renal fibrosis with decreased kidney function, have been reported due to dysbiosis [3,4]. Gut dysbiosis also affects the immune system activating regulatory T and T-helper cells, increasing the production of interferon and many other cytokines. Also, gut dysbiosis in patients with CKD causes mitochondrial dysfunction due to the high production of reactive oxygen species (ROS) in cells, contributing to the loss of kidney function [5].

Gut dysbiosis in patients with CKD leads to complications, such as increased uremic toxins and reduced SCFA production, causing loss of residual kidney function. However, the link between gut dysbiosis and kidney function is bidirectional since uraemia can also lead to gut microbiota imbalance due to high urea secretion into the intestine lumen, which is hydrolysed by urease from bacteria, forming ammonia and other toxic products, like ammonium hydroxide that cause damage to the intestinal mucosa, pH and also disturb the commensal bacterial environment [6,7,8]. In addition, patients with CKD use several medications and present many comorbidities, including diabetes mellitus, contributing to gut dysbiosis [8,9].

Modulating gut microbiota composition is a promising therapeutic strategy to reduce complications in the toxic uremic *milieu*. As diet is the primary driver of altered gut microbiota, dietary changes are the best choice to modulate the microbiota. These include more dietary fibres and bioactive compounds and reduced animal protein intake (mainly red meat) by changing to a plant-based dietary pattern. Although some controversies exist, supplementation with probiotics, prebiotics and synbiotics is a valid option [10]. It is crucial to notice that the loss of residual kidney function is closely associated with high uremic toxin plasma levels, and gut microbiota production can be a secondary cause that aggravates the accumulation of uremic toxins. This could cause a vicious circle. Thus, preservation of the residual kidney function and modulation of the gut microbiota are complementary strategies to mitigate the malefic consequences of dysbiosis [11]. This narrative review discusses the role of gut microbiota metabolism on kidney function and vice versa, as well as how we could preserve residual kidney function by targeting the gut microbiota balance. Information from Medline data was used to write this paper.

## 2. Residual Kidney Function in CKD

Preventing progression to kidney failure is an essential outcome for nondialysis CKD patients. Still, residual renal function is also crucial in preventing morbidity and mortality for patients undergoing dialysis. Patients with residual renal function have high clearance of uremic toxins, hydro-electrolytic balance, lower inflammation, improved anaemia control and a better quality of life [12].

While blood pressure control and “disease-modifying” agents, such as renin–angiotensin–aldosterone system (RAAS) inhibitors and sodium-glucose cotransporter-2 (SGLT2) inhibitors, reduce the risk of kidney failure, dietary strategies to prevent CKD progression are a current patient-prioritised area of research inquiry [13]. The Kidney Disease Outcomes Quality Initiative (KDOQI) guidelines recommend low protein diets for patients with stages 1–5 CKD before dialysis to reduce progression to kidney failure [14]. A low-protein diet reduces glomerular hyperfiltration and interstitial fibrosis [15]. In addition to favourable effects on gut microbiota, plant-based diets, in particular, have several potential health benefits, including reducing blood pressure, acidosis and hyperphosphatemia, which may reduce the risk of progression of CKD [16]. A patient-centred, plant-dominant and low-protein diet (PLADO) may not only prevent/delay dialysis initiation but may also be a key strategy for conservative kidney management [15].

For haemodialysis (HD) patients, better residual kidney function is associated with better survival and quality of life outcomes. This is because of improved volume and electrolyte management, reduced inflammation and greater clearance of middle molecules [17]. Unfortunately, timed urine samples to calculate kidney clearance of urea/creatinine are only routinely collected in some regions. However, as a surrogate, self-reported urine volume is associated with lower mortality risk [18]. In patients undergoing peritoneal dialysis (PD), residual kidney clearance confers survival benefits over comparable dialytic small solute clearance [19]. In a re-analysis of Canada–USA (CANUSA) Peritoneal Dialysis Study, for every residual kidney GFR of 5 L/week/1.73 m^2^, there was a 12% reduction in risk for mortality. In terms of urine volume, for every 250 mL increase in urine volume, there was a 36% reduction in risk for mortality [20]. Preservation of residual kidney function has also been associated with improved PD technique survival [21]. PD offer benefits over HD as a continuous dialysis modality for preserving residual kidney function in incident dialysis patients [22]. Indeed, dialysis-induced systemic stress and multiorgan injury from repeated episodes of hypovolemia and intradialytic hypotension are common complications of HD that can cause end-organ damage [23]. It should be stated that overhydration does not preserve residual kidney function [24], and hypervolemia also causes end-organ damage. There is emerging evidence that gut microbial composition may be altered by volume status; in PD patients, *Akkermansia* has been positively correlated with overhydration [25].

General strategies to preserve residual kidney function in the setting of CKD include avoidance of nephrotoxic exposures (e.g., nonsteroidal anti-inflammatory drugs and aminoglycosides); avoiding dehydration and repetitive episodes of acute kidney injury; blood pressure control with avoidance of intradialytic hypotension; and the use of incremental/individualised dialysis doses [17,19]. Higher dietary protein intake is generally recommended for patients on dialysis compared with non-dialysis CKD to account for protein losses during dialysis. However, a few studies suggest that in selected dialysis patients (e.g., those without significant malnutrition/inflammation and/or those initiating incremental HD), a low-protein diet combined with keto-analogues preserves kidney function [26,27]. The effectiveness of applying precision medicine/precision nutrition treatments to delay the progression of kidney disease is an important area of future research. Although gut dysbiosis may be related to residual renal function, the bidirectional relationship between these crucial features has been poorly investigated.

## 3. Gut Microbiota

The human body is colonised by many microorganisms, including bacteria, archaea, viruses, and fungi [28] that coexist. It is being extensively studied because of presenting the highest amount and diversity of microorganisms in the body, with more than 100 trillion microorganisms living in symbiosis [29]. Different microbiota compositions exist among the various human body niches, including oral, gastrointestinal, skin, and vagina [30]. Recently, the gut microbiota has gained mounting attention in renal research. Despite each individual having a unique gut microbiota profile, the phyla *Firmicutes* and *Bacteroidetes* are the most abundant [29,31]. Moreover, *Clostridium*, *Lactobacillus*, *Bacillus*, *Ruminococcus*, and *Enterococcus* are the most frequent genera inside the *Firmicutes* phylum [32]. The interindividual variation and flexibility in the gut microbiota have been inhibiting the definition of “healthy” gut microbiota. Several factors can affect its composition, such as birthing and infant feeding methods, age, diet, air pollution, microplastics, medications, and diseases [33].

The gut microbiota plays essential functions in the host body, including nutritional status, energy metabolism, metabolic, protective structure, immunity, and others [33,34]. Beyond these functions, the gut microbiota can have a systemic effect through its metabolites produced, such as short-chain fatty acid (SCFA), neurotransmitters, trimethylamine (TMA), and others [32]. However, these metabolites depend on the gut microbiota balance composition (dysbiosis or eubiosis). Dysbiosis, a broad term, can be described as an imbalance of the gut microbiota, compositional and functional, leading to an unhealthy event [35,36]. Bacterial richness and evenness, known as α-diversity, are associated with a healthy status [37]. In this context, it is well known that the gut microbiota plays a crucial role in the health and disease of the host organism [38], including diabetes mellitus, in which gut dysbiosis is considered a hallmark in pathogenesis [9].

## 4. Gut Microbiota Composition in CKD: Is Dysbiosis a Problem?

In 1978, it was first reported that patients with CKD had an altered gastrointestinal microbiota, with an increased abundance of streptococci, *lactobacilli* and *Bacteroides*. The change was associated, at that time, with the low intestinal transit time and uremic immune dysfunction [39]. Subsequently, Hida et al. (1996) observed that patients undergoing HD presented an altered gut microbiota profile with an increased abundance of aerobic bacteria and uremic toxins producers [40]. Wang et al. (2012) analysed the intestinal microbiota of patients undergoing PD. They reported lower values of bacteria *Bifidobacterium* species, *B. catenulatum*, *B. longum*, *B. bifidum*, *L. plantarum*, *L. paracasei* and *K. pneumoniae* compared to controls [41]. Vaziri et al. (2013) observed a difference of 190 operational taxonomic units with a high abundance of Bacteroides, Actinobacteria and Firmicutes in CKD [42]. More recently, it was observed in patients with CKD an increased abundance in the phyla *Proteobacteria*, *Actinobacteria*, and *Firmicutes* [43]. Gram-negative Proteobacteria phylum, common in the uremic flora, is associated with increased gut permeability, lipopolysaccharides (LPS) translocation and inflammation [44]. The genera *Eggerthella lenta*, *Actinomyces Streptococcus*, *Clostridium III* and *Faecalicoccus* from *Actinobacteria*, and *Firmicutes* phyla are related to inflammation [45].

Patients with CKD exhibit increased intestinal aerobes and reduced anaerobic bacteria with increased production of end-products of protein fermentation [46]. Uremic patients possess an increased number of bacterial genera possessing urease, uricase, indole- and p-cresol-forming enzymes, associated with higher production of uremic toxins [47]. Recent studies reinforce this data and show that beyond the increase in the genera of bacteria related to the production of uremic toxins, patients with CKD have a reduction in the genera of bacteria that produces SCFA [48,49,50]. SCFAs have several beneficial effects, such as improving the intestinal permeability, reduction of inflammation and oxidative stress [51,52]. The bidirectional kidney–gut axis needs more attention in CKD, in whom the gut microbiota composition is affected by many factors, such as uraemia per se, fibre dietary restriction, antibiotics use, slow colonic time, oral iron and phosphate binders, contributing to developing a dysbiotic environment [38].

Dysbiosis in CKD is commonly associated with inflammation and oxidative stress due to increased intestinal permeability, leading to endotoxin and uremic toxins translocation to the blood system [50,53]. Lipopolysaccharide is a component of the membrane of Gram-negative bacteria from the gut microbiota, a well-known bacterial endotoxin [54], which stimulates inflammatory cells with the release of proinflammatory cytokines and increases the expression of CD14 [55]. Moreover, uremic toxins, such as IS, p-CS and TMAO, promote the progression of CKD and cardiovascular disease (CVD) because they induce the production of reactive species of oxygen (ROS), increasing oxidative stress [56]. Thus, uremic gut dysbiosis in CKD patients can be a potential risk factor for CVD and loss of residual kidney function [57,58].

## 5. Effects of Gut Dysbiosis on Kidney Function

Gut dysbiosis provokes hypoxia, causing damage in the gut epithelium, since pathogenic bacteria do not contribute to SCFA production, leading to less activation of hypoxia-inducible factor-1 (HIF)—a factor essential to transcribing mucin-1 proteins that protect the integrity of the gut barrier [59]. Pathogen-associated molecular pattern (PAMP), produced during gut dysbiosis as N-formyl peptides, are associated with inflammation, e.g., *E. coli* produces fMet-Leu-Phe (fMLF), and *S. aureus* produces fMet-Ile-Phe-Leu (fMIFL) that serve as agonists of formyl peptide receptor 1 (FPR1) in neutrophils, eosinophils, monocytes, mast cells, and macrophages, including in renal tubular epithelial cells [60,61]. These receptors activate pro-inflammatory cascades through NADPH-oxidase 4 (NOX-4) stimulation, production of superoxide radicals and increased expression of nuclear factor-κB (NF-κB) [59]. A study using selective inhibitors for FPR1 and Fpr1-deficient mice showed that FPR1 binding fMLF activates platelets [62].

As discussed, uremic gut dysbiosis promotes an imbalance between proteolytic and saccharolytic bacteria. Increased urease activity by proteolytic communities leads to high syntheses of uremic toxins from amino acid fermentation. These toxic bacterial metabolites and bacteria are translocated into the circulation since there is a reduced integrity of the gut barrier, causing immune system activation. The Toll-like receptor-4 (TLR-4) can be activated in the immune cells in the kidneys by LPS and peptidoglycan, which binds the receptor, leading to stimulation of NF-κB and NLRP3 inflammasome, responsible for the synthesis of many inflammatory cytokines closely linked to the progression of kidney failure [63]. Additionally, uremic toxins promote kidney function deterioration and progression [64,65]. These toxins can damage kidney function indirectly, contributing to inflammation and oxidative stress systemic or directly, causing damage in situ in the kidney tubular cells. Uremic toxins activate aryl hydrocarbon receptor (AhR) and p53 leading to kidney fibrosis, podocyte damage and albuminuria due to transforming growth factor beta (TGF-β) activation [65,66,67].

Indoxyl sulfate produced by gut microbiota from tryptophan dietary fermentation activates the aryl hydrocarbon receptor (AhR) and NF-κB in podocytes and proximal tubules that lead to inflammation, glomerular injury, fibrosis and proteinuria [68,69]. Indoxyl sulfate can also activate the renin–angiotensin system and reduce klotho expression (an anti-ageing gene), contributing to renal fibrosis and loss of kidney function [8].

P-cresyl sulfate activates TWEAK-Fn14 and NF-kB in human proximal tubular epithelial cells [70]. In endothelial cells, p-cresyl sulfate causes senescence and damage, contributing to cardiovascular disease in patients with CKD [8]. Indol-3-acetic acid (IAA) can also activate AhR, increasing ROS and activating Cox-2 [71]. Benzoic acid fermented by the microbiota increases the production of hippuric acid (HA), and a study in human renal tubular epithelial cells treated with HA showed high production of ROS via stimulation of TGFβ, causing renal fibrosis [72].

Via fermentation of choline, L-carnitine and phosphatidylcholine, TMAO is produced, which is well known to accelerate kidney damage. Fang et al. (2021) reported that a streptozotocin-induced diabetes rat model that received TMAO in the drinking water not only presented inflammation via NLRP3 inflammasome activation and production of IL-1β and IL-18 but also developed kidney fibrosis [73].

All of these uremic toxins are involved in mitochondrial dysfunction since it causes an alteration in proteins involved with mitofission, leading to reduced mitochondrial mass and biogenesis and high production of ROS due to a disturbance in electron transport involving NADPH oxidase. Activation of the renin–angiotensin–aldosterone system and inflammation provoked by high uremic toxin levels is also associated with mitochondrial dysfunction [74]. As previously discussed by our group, several metabolites produced by the gut microbiota, such as LPS, uremic toxins and hydrogen sulfide (from the degradation of sulfur amino acids by bacteria), can alter the mitochondria function and, consequently, cause kidney damage [75].

Specific bacterial genera have been associated with immune system alteration and inflammation. *Escherichia coli* harms kidney function in cardiac surgery-associated acute kidney injury since they can metabolise phenylalanine and tryptophan (aromatic amino acids) [76]. Indole producers, such as *Escherichia coli* and *Enterobacteriaceae*, are increased in stage 5 CKD [77].

In summary, the gut microbiota interacts with kidney function; in turn, renal systems can cause alteration in the gut microbiota. Interventions to modulate the gut microbiota in CKD are beneficial since they reduce uremic toxicity, oxidative stress, mitochondrial dysfunction, and inflammation. Reduced uremic toxins and LPS production may reduce kidney failure and slow CKD progression [64]. Figure 1 shows the effects of gut dysbiosis on kidney damage and loss of kidney function.

## 6. Gut Microbiota Modulation and Effects on Residual Kidney Function

Modulating the gut microbiota in CKD by blocking or reducing the production of uremic toxins may delay kidney disease progression. We present recent studies showing strategies that could slow the loss of kidney function and preserve residual kidney function. The modulation of gut microbiota composition may mitigate chronic kidney damage, and diet has gained increasing attention in recent years since it is the primary driver of the gut microbiota. Also, it is important noticing that modulation of gut microbiota can improve the metabolism in patients with diabetes mellitus, a significant cause of kidney damage progression [9].

### 6.1. Fibres

The fact that uremic toxins promote a progressive decline in renal function is well recognised [78]. Dietary fibre intake decreases uremic toxins plasma levels through favouring saccharolytic bacteria; improving intestinal motility and, consequently, decreasing toxins absorption; and increasing SCFA production, which contributes to the maintenance of the colonic epithelium integrity [16]. Resistant starch (RS), resistant oligosaccharides, fructooligosaccharides (FOS), galactooligosaccharides (GOS), pectins, inulin, dextrin, glucomannan and gums are all examples of fermentable fibres [79]. The SCFAs are a communication pathway among diet, gut microbiota and host cells, and a relationship exists between plasma SCFA and kidney function. A study involving diabetic and nondiabetic individuals, using Mendelian randomisation, showed that valerate plasma levels, an SCFA, were positively associated with eGFR in the total sample and among nondiabetic subjects but not in nondiabetics [80]. A longitudinal study of stage 3–4 CKD patients showed that the group that had a higher dietary fibre intake (≥25 g/day) had a slower decrease in the eGFR over the 18-month follow-up period and lower levels of proinflammatory factors, IS and serum cholesterol [81].

Ebrahim et al. (2022) investigated the effect of ß-glucan supplement (6 g of fibre, 3 g were ß-glucan) on kidney function, uremic toxins plasma levels and gut microbiota in CKD 3–5 patients. IS, p-CS and p-cresyl glucuronide (pCG) plasma levels improved with the intervention without any change in kidney function over the 14-week study period. Furthermore, the supplementation beneficially affected the gut microbiome. It can be speculated that when the fibre supply is sufficient, gut microbiota uses less protein as an energy source, resulting in lower uremic toxin production [82].

A meta-analysis of five randomised clinical trials and 179 HD patients showed that resistant starch (RS) improves residual kidney function. The hypothesis is that modulation of gut microbiota with RS increases the beneficial bacteria taxa and reduces the production of uremic toxins associated with less cytokine production, improving residual kidney function [83]. Studies in nephrectomised rats have documented the role of RS on kidney function, as crude potato starch increased SCFA production and reduced urea levels [84]. Subsequent experimental studies showed that RS supplementation mitigates renal histological damage and improves kidney function by modulating the microbiota and the transcription factor Nrf2, which has antioxidant effects [78,79,80]. The effects of fibres on kidney function vary. The type of fibre supplement, dosage, duration of intervention, food pattern and the characteristics of the patients and their microbiota can influence the results of studies. More randomised clinical trials are necessary to reach evidence that consolidates a clinical recommendation [85].

### 6.2. Probiotics

Probiotics have emerged as a potential adjuvant intervention to restore the gut microbiome in CKD through bacteriocin production; increasing intestinal motility and, consequently, reducing the exposure time to toxic metabolites; attenuating increased intestinal permeability; modulating the host’s innate and adaptive immune systems; and modulating gut-derived uremic toxins and SCFA production. As a result, probiotics may contribute to mitigating inflammation, oxidative stress and, finally, the progression of CKD [86]. Several studies have explored this promising intervention. The effects of fortified soy milk with *Lactobacillus plantarum* A7 on kidney function were assessed by Miraghajani et al. (2019) in patients with diabetic nephropathy (stages 1–2). It was shown that eight weeks of intervention (200 mL/day probiotic soy milk) resulted in a significant reduction of kidney function biomarkers cystatin C and progranulin compared to the control group (soy milk) [87]. However, it should be noted that the study subjects had well-preserved kidney function at baseline.

In contrast, we showed that three months of supplementation with a combination of three probiotic strains (9 × 10^10^ UFC *Streptococcus thermophilus* KB19, *Lactobacillus acidophilus* KB27 and *Bifidobacteria longum* KB31) did not change eGFR or uremic toxins plasma levels in CKD 3–5 patients. Instead, the patients exhibited a significant increase in IL-6 levels [88,89].

Recently, Zhu et al. (2021) showed that the *Lactobacillus casei*, a probiotic bacteria isolated from Chinese fermented sour kinds of milk, slows the progression of acute and chronic kidney disease in a mouse model of bilateral ischaemia–reperfusion injury. The effects were associated with increased levels of SCFA and nicotinamide in serum and kidney, resulting in modulation of inflammatory response and reduced kidney damage. In addition, they demonstrated that three months of treatment with the same probiotic (4 × 10^9^ CFU per day, in sachets) was well tolerated and slowed the decline of kidney function measured by serum cystatin C [90].

Fermented foods, such as dairy, fruits and vegetable-based fermented food, contain bacteria with probable probiotic effects [91]. Experimental studies show that fermented foods may mitigate many complications associated with CKD progression and gut dysbiosis [92,93,94]. It is likely that salutogenic bioactive compounds naturally present in foods or those derived from the fermentation process (bactericins, exopolysaccharides and SCFA, among others) also contribute. A meta-analysis of ten randomised controlled trials involving 359 patients with CKD evaluated the effects of the probiotic intervention on kidney disease progression. It was shown that the urea level was significantly reduced in nondialysis patients receiving probiotics. However, no significant impact of probiotic supplements on uric acid, C-reactive protein, creatinine and eGFR was found [95].

Although probiotics may have a salutary potential on the biochemical uremic *milieu*, the usefulness of probiotics in delaying the progression of kidney failure has yet to be confirmed [95,96]. It is essential to highlight some factors, such as the heterogeneity of the doses, strains and treatment time used in the studies. The creatinine-based eGFR used in many studies to measure the effects of probiotics on renal function is questionable because probiotics with creatininase activity can increase the extrarenal clearance of creatinine, decreasing the creatinine concentration in the plasma [96]. Finally, probiotic effects might be affected by gut environment impairment by specific disease-related factors [89,97]. As already mentioned, diet is the main factor influencing gut microbiota’s structure and metabolic activity. Diet shapes our gut microbiome by modulating the abundance of specific species and their individual or collective functions [98]. In addition to probiotic intervention, when aiming to modulate the gut microbiota, a background diet rich in fruits, vegetables, and high-fibre products is, therefore, essential [99].

### 6.3. Synbiotics

In 2019, the International Scientific Association for Probiotics and Prebiotics (ISAPP) updated the synbiotic definition as “a mixture comprising live microorganisms and substrate(s) selectively utilised by host microorganisms that confers a health benefit on the host” [100]. Thus, synbiotics have emerged as a potential therapeutic strategy to modulate the gut microbiota and, hence, decrease harmful metabolites, such as uremic toxins, related to uremic dysbiosis [2]. One of the first studies regarding the effects of synbiotics for patients with CKD observed that one packet of synbiotics three times a day for two weeks improved constipation and reduced p-CS serum levels in patients undergoing HD [101]. Another study using a commercial synbiotic found a significant reduction in p-CS levels in CKD 3–4 [102]. Similarly, six weeks of synbiotic intervention in nondialysis CKD patients with 15 g/day of prebiotic (inulin, fructooligosaccharides and galactooligosaccharides) plus two capsules with 45 billion CFU/day (*Lactobacillus*, *Bifidobacteria* and *Streptococcus*) resulted in a reduction in p-CS plasma levels. A change in the microbiota composition, with an increase in the *Bifidobacterium* and depletion of Ruminococcaceae was also observed [103]. In a follow-up study with 12 months of intervention time in moderate to severe CKD, they used a different supplementation with 20 g of high-resistant starch fibre in combination with 4.5 × 10^11^ CFU/day (Bifidobacteria, Lactobacillus and Streptococcus; Swiss Mendes) in a sachet. The synbiotic group presented an increase in the *Bifidobacterium* and *Blautia* spp. However, it was also found in the synbiotic group a decrease in eGRF, which was not expected and opposite from previous studies [104].

Cosola et al. (2021) observed that patients in early CKD stages using a commercial synbiotic for two months reduced serum-free IS, small intestine permeability, abdominal pain and constipation syndromes [105]. On the other hand, Lydia et al. (2022) did not find any difference in the serum levels of IS in HD patients who used a synbiotic intervention (Lactobacillus acidophilus and Bifidobacterium longum 5 × 10^9^ CFU and 60 mg of fructooligosaccharides—FOS) daily for 60 days. However, they reported significant improvement in constipation and better quality of life [106]. In addition, a recent meta-analysis study of HD patients showed that synbiotic treatment reduced p-CS plasma and had anti-inflammatory effects with reduced endotoxins levels [107]. The inconsistent findings may be due to the prebiotic or probiotic used, the dosage and the intervention time. Despite the discrepant, synbiotics seem to act on the modulation of the gut microbiota and have a beneficial effect in reducing some uremic toxins production on CKD. However, its effect on delaying progression is scarce and needs additional studies.

### 6.4. Dietary Proteins

Metabolic and hemodynamic effects of proteins are well known; thus, dietary protein restriction has long been used to slow the progression of CKD. Additionally, the quality of proteins plays a role in gut microbiota composition and function. Protein residues that escape the digestive process reaching the gut microbiota in the colon, can suffer proteolytic fermentation generating uremic toxins [108,109,110]. The amino acids phenylalanine and tyrosine are precursors of metabolites belonging to the classes of hippurates and phenols. Tryptophan participates in the formation of indoles, while ornithine, lysine and arginine lead to the formation of polyamines. These metabolites cause damage to the health of the host, including the progression of CKD [109]. In addition, dietary protein increases gut bacterial production of hydrogen sulfide (H_2_S). In a mouse model of CKD, high sulfur amino acid-containing diet resulted in post-translationally modified microbial tryptophanase activity. This reduced uremic toxin-producing activity and ameliorated the progression of CKD [111]. Moreover, ischemia/reperfusion-mediated myocardial injury and apoptosis were partially reversed by a methionine-restricted diet [112]. Based on the premise of the possible effects of dietary restriction of sulfur-containing amino acids in the protection of ischemia/reperfusion injury, a randomised clinical trial aimed to evaluate the protective impact of restricting sulfur-containing amino acids intake before cardiac surgery on the incidence of acute kidney injury (AKI). However, no beneficial effects of sulfur-containing amino acids restriction on the rate of AKI after surgery was observed [113]. The protein source, the content of other components in the diet, processing-related factors, protein oxidation and glycation, digestibility and the bioavailability of dietary proteins can influence protein fermentation and the effects on gut microbiota [110].

### 6.5. Bioactive Compounds

Polyphenols have anti-inflammatory and antioxidant properties that can be renoprotective and affect gut microbiota since they can act as a prebiotic [114]. Polyphenols usually have poor intestinal absorption, but after the biotransformation by the microbiota, they are hydrolysed and absorbed. The produced compounds benefit the intestinal cells and microbiota and promote the growth of specific bacteria [115]. Bioactive peptides from fish (e.g., monkfish) can also modulate the gut microbiota composition [116] and are linked with antioxidants, anti-inflammatory, anti-diabetic and renoprotective effects [116]. So far, experimental and clinical studies evaluating the effects of food or bioactive compounds on gut microbiota composition and kidney function are scarce (Table 1). Foods rich in bioactive compounds, such as turmeric, prebiotic (resistant starch), blueberry and cranberry, chocolate, propolis, beetroot, broccoli, garlic, cinnamon, coffee and Brazil nut, may modify the microbiota composition, modulate the transcription factors involved with inflammation and oxidative stress and could preserve kidney function [2,10,97]. Although some studies have been performed with bioactive compounds, information regarding their role in the gut microbiota composition and kidney function in CKD patients remains limited [10].

Studies have shown that garlic may alter the gut microbiota, increasing the diversity of gut microbiota composition, which is beneficial to intestinal health. Garlic can restore *Lactobacillales*, *Bifidobacterium*, *Clostridium cluster XVIII* and *Prevotella* levels [117]. Anthocyanins in fruits and vegetables can also modulate the gut microbiota composition, increasing *Bifidobacterium* spp., and *Lactobacillus-Enterococcus* spp., levels and SCFA production, improving both the immune system and intestinal barrier [118]. In a recent review, the role of berry fruits on gut microbiota modulation and on, the modulation of proteins involved with the gut barrier and the essential role in improving the uremic condition in patients with CKD was discussed [119]. Berry fruits, nuts and pomegranates contain ellagic acid and ellagitannins that the gut microbiota ferments to produce urolithins with antimicrobial properties [120].

Cinnamon compounds such as cinnamaldehyde, trans-cinnamaldehyde, cinnamic acid and cinnamyl acetate also may improve the gut barrier integrity and modulate the gut microbiota, promoting the growth of salutogenic bacteria [121]. Beetroot is known to alter the oral bacteria species, transforming NO_3−_ to NO_2−_ increasing nitric oxide (NO) synthesis. Components from beetroot increase alpha diversity and SCFA production [122]. Propolis, a resin produced by bees, has significant anti-inflammatory properties, reducing levels of cytokines as seen in HD and PD patients [123,124]. In addition, propolis has antimicrobial actions and can increase salutogenic bacterial diversity [125]. Turmeric can also change the gut microbiota composition, improve intestinal barrier permeability, and increase intestinal alkaline phosphatase catalytic activity. Patients undergoing HD curcumin supplementation for three months reduced p-cresyl sulfate [126].

The metabolites produced by the gut microbiota from polyphenols promote several benefits to the gut barrier and the immune system, increase the body’s antioxidant capacity and interact with NF-kB-related inflammation (Figure 2). The phytochemicals produced by the gut microbiota are also important in modulating the bacteria communities [127,128]. To understand the process of fermentation of polyphenols by the gut microbiota, a focused study of the gut microbial enzymes should be performed. Several mechanisms have been suggested regarding the benefits of these metabolites produced by the gut microbiota in reducing inflammation and oxidative stress in the organism systems, and these beneficial properties may extend to kidney function. However, the mechanism remains unknown.

### 6.6. Faecal Microbiota Transplantation

The faecal microbiota transplant (FMT) procedure is the transfer of the distal faecal microbiota from a healthy donor to the gastrointestinal tract of a dysbiotic recipient to re-establish the microbiota homeostasis, structure and diversity [129]. FMT can be administered through the capsule, nasogastric, nasojejunal tube or colonoscopy [130]. FMT has been described in the literature since the 4th century in China [131]. FMT leading to the recolonisation of the uremic gut with a healthy microbiome could be a valid strategy for reducing dysbiosis, increasing salutogenic bacterial richness and reducing retention of uremic toxins, LPS and pro-inflammatory mediators [132]. In this context, FMT is a well-established treatment therapy for *Clostridium difficile* infection, with a 90% success rate. In other conditions, such as autoimmune, neurodegenerative, autism, diabetes, cancer and CKD, FMT studies have demonstrated substantial benefits [133].

Wang et al. (2020) transferred the FMT of patients with CKD and healthy individuals into germ-free CKD mice or antibiotic-treated CKD rats, and they observed that the animals that received that FMT from the CKD patients increased the production of uremic toxins, oxidative stress, and interstitial fibrosis [134]. Another study found that mice that received FMT from STZ-induced DKD mice with severe proteinuria had different microbiota compositions and hence higher production of TMAO and LPS [135]. These studies demonstrate that the CKD gut microbiota might interfere with the progression or severity of the disease. Two recent studies reported the beneficial effects of FMT from healthy animals in animals with CKD. It was observed that CKD-induced animals that received the FMT changed the gut microbiota composition leading to a significant reduction in the production of the uremic toxins (cresol pathway) and improved tubulointerstitial injury [136,137]. In a recent study, Bastos et al. (2022) reported that FMT in a preclinical model of type-2 DM, obesity, and DKD using BTBR^ob/ob^ mice, an increase in the abundance of the *Odoribacteraceae* bacteria family accompanied by a reduction of albuminuria and TNF-alpha. The authors declared that FMT is a safe treatment [138].

Few studies have been conducted in humans. In a case report, Zhou et al. (2021) reported that after two months of FMT, a patient with membranous nephropathy improved the nephrotic syndrome and kidney function [139]. In another case report on 6–7 months of FMT treatment with two IgA nephropathy patients, the 24 h urinary protein decreased, and the serum albumin increased in addition to an alteration in the gut microbiota [140]. Since FMT restore the gut microbiota and mitigate the production of uremic toxins and inflammation, FMT might be a promising strategy to retard the progression of CKD.

### 6.7. Medications

A complex bidirectional interaction exists between medications and gut microbiota [141]. Although the effect of drugs on the gut microbiota has been studied over the past few years, the impact of the gut microbiota on the metabolism of drugs commonly used for CKD remains untested. Patients with CKD undergo polypharmacy and commonly use many medications, such as antibiotics, iron, and proton pump inhibitor, with a well-documented negative impact on the uremic gut microbiota [142,143,144,145]. For other drugs, such as phosphate binders and oral adsorbent AST-120, the effects on the gut microbiota are better known.

The oral adsorbent AST-120 is a carbon particle in the intestine that absorbs uremic toxins and eliminates them by faeces [146]. In a hypertensive and CKD-induced animal model, Yoshifuji et al. (2018) reported that 12 weeks of treatment with AST-120 restored gut permeability by increasing the expressions of the tight junction proteins. They also observed that AST-120 affected the genus *Lactobacillus*, which is associated with tight junctions through the Toll-like receptor 2 pathway [147]. Another animal model study found that treatment with AST-120 for seven weeks altered the gut microbiota by reducing the abundance relative to *Erysipelotrichaceae uncultured* and *Clostridium sensu stricto 1* (two genera related to the production of p-CS) [148]. Recently, in the first clinical study of CKD patients, AST-120 changed the β-diversity and increased the abundance of *Clostridium_sensu_stricto_1*, *Ruminococcus_2*, *Eubacterium_nodatum* and *Phascolarctobacterium* and decreased the abundance of the family Erysipelotrichaceae and the genus Coprococcus 3 (related to uremic toxins production) [149].

Another commonly used class of drugs in CKD are calcium and non-calcium-based phosphate binders). When Lau et al. (2018) administrated 4% of ferric citrate for six weeks in mice, they observed that whereas the α-diversity in *Clostridiaceae* and *Enterobacteriaceae* increased, uremic toxins levels did not alter. Moreover, *A. muciniphila* (from the phylum Verrucomicrobia), well known to have an anti-inflammatory effect, increased after the intervention [150]. In contrast, studies with sucroferric oxyhydroxide intervention found no substantial differences in the uremic gut microbiota [151,152,153].

## 7. Conclusions

Gut dysbiosis in patients with CKD is involved with the loss of residual kidney function since the increased uremic toxin production, and endotoxemia are associated with complications, such as inflammation, oxidative stress, mitochondrial dysfunction, senescence and endothelial damage. Clinical studies must investigate if interventions that mitigate gut dysbiosis in CKD can slow progression and maintain residual renal function via inhibiting uremic toxins. Promising strategies include modulation of gut microbiota through increased intake of fibres, fermented food, plant-based polyphenols and faecal transplantation.

## Figures and Tables

**Figure 1 toxins-15-00499-f001:**
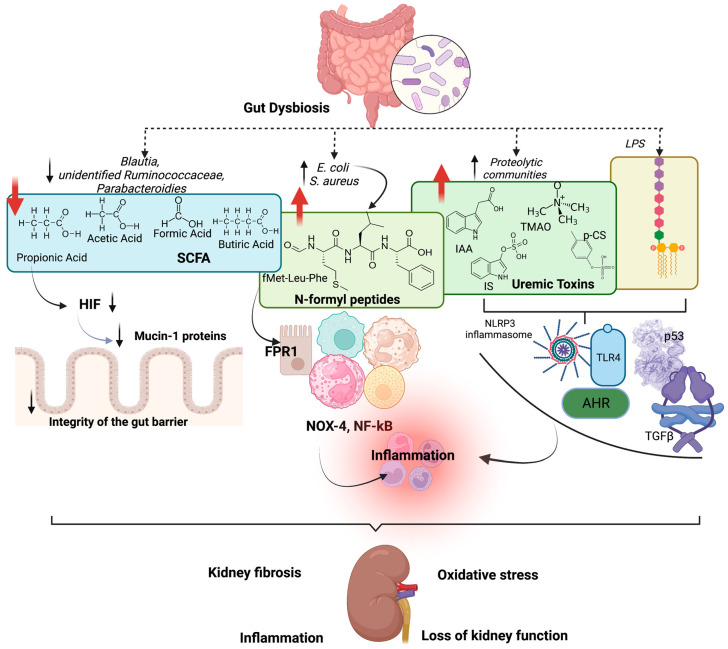
Effects of gut dysbiosis on kidney damage. Gut dysbiosis contributes to the reduction of short-chain fatty acids (SCFAs) production, provoking reduced hypoxia-inducible factor-1 (HIF) activation and, consequently, reducing mucin-1 proteins, leading to the impairment of the integrity of the gut barrier. In addition, dysbiosis leads to the formation of N-formyl peptides, such as fMet-Leu-Phe (fMLF), which activates the formyl peptide receptor 1 (FPR1) in neutrophils, eosinophils, monocytes, mast cells, and macrophages and also in renal tubular epithelial cells, stimulating NADPH-oxidase 4 (NOX-4) and nuclear factor (NF)-κB (NF-kB), increasing inflammation. Also, uremic toxins, such as indoxyl sulfate (IS), p-cresyl sulfate (p-CS), indol acetic acid (IAA) and trimethylamine-N-oxide (TMAO), together with lipopolysaccharides (LPS) activate Toll-like receptor-4 (TLR-4), NF-kB, NOX-4, aryl hydrocarbon receptor (AhR), transforming growth factor beta (TGF-β) and p53, which are associated with inflammation, oxidative stress, kidney fibrosis and CKD progression. Created by BioRender.com.

**Figure 2 toxins-15-00499-f002:**
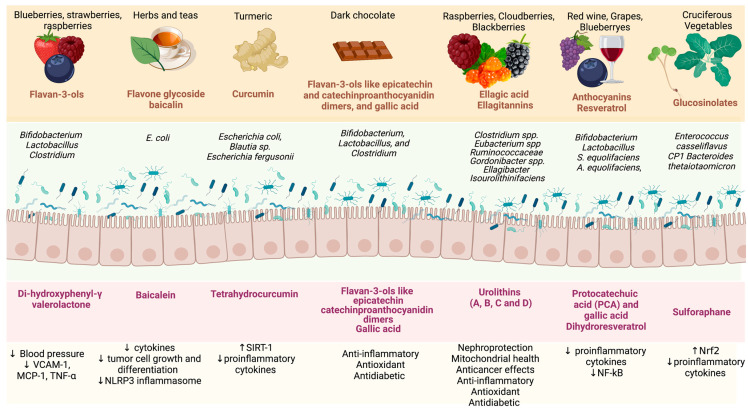
Benefits of the metabolites produced by the gut microbiota from polyphenols found in food. Abbreviations: VCAM-1: vascular cell adhesion molecule-1; TNF-∝: tumour necrosis factor; MCP-1: monocyte chemoattractant protein-1; NLRP3: NLR family pyrin domain-containing 3; SIRT-1: sirtuin-1; NF-kB: nuclear factor-κB; Nrf2: nuclear factor erythroid 2-related factor 2. Created by BioRender.com.

**Table 1 toxins-15-00499-t001:** Experimental studies involving food or bioactive compounds supplementation, gut microbiota and kidney function.

References	Sample/Design	Intervention	Results
Polyphenols			
Delgadillo-Puga et al. (2023) [114]	Male C57BL/6 mice and INS-1E cells or C2C12 myotube	Groups: control diet, HFD, HFD + 10% VFP (Vachellia farnesiana pods), or 0.5%, 1%, or 2% or VFPE (Vachellia farnesiana polyphenolic extract) for 14 weeks Cells treated with 0, 5, or 10 mg/mL of VFPE for 2 h	VFP (Vachellia farnesiana pods) or VFPE (Vachellia farnesiana polyphenolic extract: Prevented glomerular damage, insulin resistance, hepatic steatosis, ↓ Abundance of Desulfovibrionaceae, Enterobacteriales, Gram-positive bacteria Erysipelotrichales, Chloroplast, Mollicutes, and Mycoplasmatales INS-1E cells: ↓ insulin secretion C2C12 myotubes: ↑ mitochondrial activity
Ren et al. (2022) [116]	C57BL/6J mice treated with Lophius litulon peptides (LPs) isolated from Monkfish	HFD (saline), HFD + LP–100 mg/kg and HFD + LP–200 mg/kg	LP: ↓ kidney damage ↓UA, Cr, BUN ↓ IL-1β, IL-6, and TNF-α ↓ TLR4/NF-κB pathway ↑ Nrf2/Keap1 pathway ↑ out and bacterial diversity
Zou et al. (2022) [90]	Male C57BL/6 mice cisplatin-induced AKI	Control group; cisplatin group; Qiong-Yu-Gao (QYG) group	↓ gene expression of kidney injury markers ↓ BUN ↓ fibrosis induced by cisplatin and inflammation diversity of bacteria *Akkermansia*, *Faecalibaculum*, *Bifidobacterium* and *Lachnospiraceae_NK4A136* SCFA-producing bacteria

Abbreviations: HFD: high-fat diet; UA: uric acid; Cr: creatinine; BUN: blood urea nitrogen; IL-6: interleukin-6; IL-1β: interleukin-1β; TNF: tumour necrosis factor; OTU: operational taxonomic unit; SCFAs: short-chain fatty acids; AKI: acute kidney injury.

## Data Availability

Not applicable since this is a narrative review paper.

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
