# Peer review of "Gut Microbiota Interventions to Retain Residual Kidney Function"

_toxins, 2023, doi:10.3390/toxins15080499_

Round 1

Reviewer 1 Report

Very nice manuscript! You structured it good. The information is scientific and very detailed.  The info is new for the reader. You can improve it a bit. Make something like this food influences this germ and than this germ influences DKD.  You can check: Unravelling the involvement of gut microbiota in type 2 diabetes mellitus. Life Sci. 2021 May 15;273:119311. doi: 10.1016/j.lfs.2021.119311. Epub 2021 Mar 1. PMID: 33662428.

English is ok.

Author Response

Very nice manuscript! You structured it good. The information is scientific and very detailed.  The info is new for the reader. You can improve it a bit. Make something like this food influences this germ and than this germ influences DKD.  You can check: Unravelling the involvement of gut microbiota in type 2 diabetes mellitus. Life Sci. 2021 May 15;273:119311. doi: 10.1016/j.lfs.2021.119311. Epub 2021 Mar 1. PMID: 33662428.

Thank you for your suggestion. We have added this paper to our review.

Reviewer 2 Report

The review-type study, organised into seven sections, is focused on a significant area of research – gut dysbiosis in chronic kidney disease (CKD). Gut dysbiosis has been registered in CKD patients, generally manifesting as an imbalance between beneficial and pathogenic bacteria. Recently, a low sodium, potassium and phosphate diet associated with dietary fibres may be used to establish the normal gut microbiota in CKD patients.

To improve the quality of the manuscript, I recommend several suggestions to the authors:

Major comment:

-           Please expand the data regarding the relationship between microbiota and CKD in the Introduction section (Cao et al. 2022, Doi: 10.3389/fmed.2022.829349;  Tourountzis et al., 2022 Doi: 10.3390/life12101513; Wehedy et al., 2022 DOI: 10.3389/fmed.2021.790783)

-           The molecular mechanisms of gut dysbiosis contributing to the pathogenesis and progression of CKD are incompletely described, especially the gut microbiota-mitochondria axis (please see lines 188-196);

-           Kindly add the limits and the perspective of this study.

Minor comments:

-           Kindly add a reference related to data shown in lines 23, 331, and 388;

-           All the abbreviations used in the main text should be explained (e.g. KDOQI in line 57)

-           Please explain in the Figure 2 legend all the abbreviations used in this Figure.

-           Kindly revise the references list according to the Toxins journal recommendation (e.g. the Journal name should be written in Italics).

Kindly revise the following sentences:

Lines 21-23 - Although most studies have been conducted in hemodialysis patients, a few studies have been conducted in patients on peritoneal dialysis and stages 1-5 before dialysis.

Lines 101-102 - Different microbiota exists in the human body including the oral, gastrointestinal, skin, vagina, and gut microbiota.

Lines 259-262 - Subsequent experimental studies showed that RS supplementation mitigate renal  histological damage and improve kidney function, modulating the microbiota and the  transcription factor Nrf2 involved with the antioxidant synthesis as reducing the expression of NF-kB, involved with cytokines synthesis

Author Response

Please expand the data regarding the relationship between microbiota and CKD in the Introduction section (Cao et al. 2022, Doi: 10.3389/fmed.2022.829349; Tourountzis et al., 2022 Doi: 10.3390/life12101513; Wehedy et al., 2022 DOI: 10.3389/fmed.2021.790783).

Thank you for your suggestions. We have added these papers in the introduction.

The molecular mechanisms of gut dysbiosis contributing to the pathogenesis and progression of CKD are incompletely described, especially the gut microbiota-mitochondria axis (please see lines 188-196);

Thank you for your comment. We have described more mechanisms and also added information about the link between gut dysbiosis and mitochondrial dysfunction.  

Kindly add the limits and the perspective of this study.

Although various studies have shown the influential role of diet in modulating gut microbiota composition, the effects of this modulation on residual kidney function remain limited. With this review, we discussed the role of gut microbiota metabolism on residual kidney function and how the residual kidney function could be preserved by modulating the gut microbiota.

Minor comments:

Kindly add a reference related to data shown in lines 23, 331, and 388;

The refs were added.

All the abbreviations used in the main text should be explained (e.g. KDOQI in line 57)

Thank you. We have corrected them.

Please explain in the Figure 2 legend all the abbreviations used in this Figure.

We have added the abbreviation. Thank you.

Kindly revise the references list according to the Toxins journal recommendation (e.g. the Journal name should be written in Italics).

We have corrected them.

Comments on the Quality of English Language

Kindly revise the following sentences:

Lines 21-23 - Although most studies have been conducted in hemodialysis patients, a few studies have been conducted in patients on peritoneal dialysis and stages 1-5 before dialysis.

Lines 101-102 - Different microbiota exists in the human body including the oral, gastrointestinal, skin, vagina, and gut microbiota.

Lines 259-262 - Subsequent experimental studies showed that RS supplementation mitigate renal  histological damage and improve kidney function, modulating the microbiota and the  transcription factor Nrf2 involved with the antioxidant synthesis as reducing the expression of NF-kB, involved with cytokines synthesis.

Thank you. We have corrected them.

Reviewer 3 Report

Overall, this review publication offers a good overview on the interactions between gut microbiota and kidney physiology and disorders. This manuscript shows rich content, providing a deep insight for some works. The study is within the journal’s scope, and I found it to be well-written, providing sufficient information. However, before publication some points need to be clarified.

My comments:

Line 19 – I do not understand the idea of Introduction in review article. Where are results and discussion, then?

Line 46 – please present a goal of this review and add a short methodology of it.

Line 57 – please expand KDOQI acronym.

Line 72 – please explain what CANUSA stands for.

Line 125 – the autohrs should follow journal references style guidelines. In the whole manuscript, please avoid sentence contruction like “Hida et al. (1996) observed”. Authors names should be replaced with appropriate reference numbers.

Line 155 – the authors should ensure that they use the term “expression” in relation to genes only.

Line 303 – in the main text CRP acronym is used only one time. I see no sense to abbreviate it.  

Line 394 – the bacteria phyla names like “Lactobacillales, Bifidobacterium, Clostridium etc.” should be written in italics.

Line 498 – conclusions are too laconic. Please provide also some future perspectives.

Author Response

Line 19 – I do not understand the idea of Introduction in review article. Where are results and discussion, then?

This is a narrative Review and usually, in this type of paper, we add an introduction to set the scene, explain the context to the reader and give the reader a background of the review and the motivations that led to it. Also, narrative reviews have a conclusion to recapitulate the main points discussed.

Line 46 – please present a goal of this review and add a short methodology of it.

The aim is described at the end of the introduction: This narrative review discusses the role of gut microbiota metabolism on kidney function and vice-versa and how we could preserve residual kidney function by targeting the gut microbiota balance.

We have described a short method as requested.

Line 57 – please expand KDOQI acronym.

Line 72 – please explain what CANUSA stands for.

Thank you. We have corrected them.

Line 125 – the autohrs should follow journal references style guidelines. In the whole manuscript, please avoid sentence contruction like “Hida et al. (1996) observed”. Authors names should be replaced with appropriate reference numbers.

We have added the number of this ref.

Line 155 – the authors should ensure that they use the term “expression” in relation to genes only.

Line 303 – in the main text CRP acronym is used only one time. I see no sense to abbreviate it.  

Line 394 – the bacteria phyla names like “Lactobacillales, Bifidobacterium, Clostridium etc.” should be written in italics.

Thank you for your suggestions.

Line 498 – conclusions are too laconic. Please provide also some future perspectives.

We have added more information to the conclusion.

Round 2

Reviewer 2 Report

The manuscript has been improved by responding to comments and integrating missing references.